# The Current Knowledge on *Clostridioides*
*difficile* Infection in Patients with Inflammatory Bowel Diseases

**DOI:** 10.3390/pathogens11070819

**Published:** 2022-07-21

**Authors:** Alina Boeriu, Adina Roman, Crina Fofiu, Daniela Dobru

**Affiliations:** 1Gastroenterology Department, University of Medicine Pharmacy, Sciences, and Technology “George Emil Palade” Targu Mures, 540142 Targu Mures, Romania; aboeriu@gmail.com (A.B.); crinafofiu@gmail.com (C.F.); danidobru@gmail.com (D.D.); 2Gastroenterology Department, Mures County Clinical Hospital, 540103 Targu Mures, Romania

**Keywords:** inflammatory bowel disease, *Clostridioides difficile*, ulcerative colitis, Crohn’s disease

## Abstract

*Clostridioides difficile* (*C. difficile*) represents a major health burden with substantial economic and clinical impact. Patients with inflammatory bowel diseases (IBD) were identified as a risk category for *Clostridioides difficile* infection (CDI). In addition to traditional risk factors for *C. difficile* acquisition, IBD-specific risk factors such as immunosuppression, severity and extension of the inflammatory disease were identified. *C. difficile* virulence factors, represented by both toxins A and B, induce the damage of the intestinal mucosa and vascular changes, and promote the inflammatory host response. Given the potential life-threatening complications, early diagnostic and therapeutic interventions are required. The screening for CDI is recommended in IBD exacerbations, and the diagnostic algorithm consists of clinical evaluation, enzyme immunoassays (EIAs) or nucleic acid amplification tests (NAATs). An increased length of hospitalization, increased colectomy rate and mortality are the consequences of concurrent CDI in IBD patients. Selection of CD strains of higher virulence, antibiotic resistance, and the increasing rate of recurrent infections make the management of CDI in IBD more challenging. An individualized therapeutic approach is recommended to control CDI as well as IBD flare. Novel therapeutic strategies have been developed in recent years in order to manage severe, refractory or recurrent CDI. In this article, we aim to review the current evidence in the field of CDI in patients with underlying IBD, pointing to pathogenic mechanisms, risk factors for infection, diagnostic steps, clinical impact and outcomes, and specific management.

## 1. Introduction

The healthcare burden of *Clostridioides difficile* infection (CDI), especially in patients with underlying chronic disorders, dictates an increased vigilance for prompt diagnosis, appropriate treatment, and prevention measures to limit the spread. The economic and clinical impact of CDI is significant. Annual costs have been estimated to exceed $1.2 billion in the United States according to various reports [1], with increased length of hospitalization and hospital costs attributed to infection [2,3,4].

Given the significant health impact in the United States, *Clostridioides difficile* (*C. difficile*) has been recognized as one of the three “urgent” antibiotic resistance threats by the Centers for Disease Control and Prevention (CDC) [5]. The selection of strains of higher virulence (ribotype 027, ribotype 078) [6,7,8,9], the emergence of antibiotic resistance [10], and recurrent infections are major health threats. The risk of CDI recurrence after the primary episode is 25%, and the risk for subsequent recurrences further increases [11].

Inflammatory bowel diseases (IBD) represent chronic disorders with a complex pathogenesis involving environmental risk factors, genetic susceptibility, dysregulation of immune response to altered gut microbiome, and disruption of the epithelial barrier [12]. Inflammation of the gastrointestinal tract may have an unpredictable evolution, with periods of remissions and exacerbations. Nonsteroidal anti-inflammatory drugs and antibiotic use, infections [13,14], smoking [15,16], psychological stress [17,18,19,20], high altitude journeys and flights [21] have all been described as triggers for ulcerative colitis (UC) or Crohn’s disease (CD) flare-ups. By examining the stool during IBD relapses, *C. difficile* toxins were detected in 5.5% to 20% of patients [22,23].

Making the distinction between symptoms related to CDI and symptoms associated with an IBD flare can be a difficult task in practice. During IBD flares, patients experience abdominal pain, bloody diarrhea, and weight loss. Similarly, watery diarrhea, abdominal cramps, dehydration, loss of appetite and weight loss, and even rectal bleeding in severe cases, are typical symptoms of CDI. According to international guidelines, patients presenting with more than three episodes of unexplained unformed stools in 24 h should be tested for CDI [24]. Therefore, patients presenting with an IBD exacerbation should be screened for *C. difficile* toxins [25,26,27].

The risk factors for *C. difficile* acquisition, complex disease, and for recurrence, are older age (greater than 65 years old), current or recent exposure to broad-spectrum antibiotics (broad-spectrum penicillins, fluoroquinolones, cephalosporins, clindamycin), hospitalization [28,29,30], comorbidities, hypoalbuminemia, renal insufficiency [31], immunosuppression [32], chemotherapy [33,34,35,36], proton pump inhibitor (PPI) usage [37,38], nasogastric tubes [39], and gastrointestinal surgery [40]. Elderly patients, patients with immunosuppression, oncology patients, solid organ transplant recipients [41], and IBD patients also represent categories susceptible to infection.

In this article, we review the current evidence on CDI in the setting of IBD and highlight epidemiological data, pathogenic mechanisms, risk factors, and clinical outcomes. Diagnostic algorithms and management strategies according to international guidelines are presented. Features specific to the CDI-IBD population are described, as well as divergent data and opinions regarding risk factors and therapeutic approaches.

## 2. Epidemiology of CDI in IBD Patients

Epidemiological data prove that the incidence of CDI among IBD patients has been increasing [42,43,44], with negative consequences on clinical outcomes (higher hospitalization rate, requirement for colectomy, and mortality). In a retrospective cohort study by Rodemann et al., the rate of CDI approximately doubled in CD patients (9.5 to 22.3/1000 admissions) and tripled in UC patients (18.4 to 57.6/1000) over a period of seven years (1998–2004) [45]. The increasing trend of CDI rate in IBD-hospitalized patients in the USA was also demonstrated in an analysis of data from the Nationwide Inpatient Sample: a 1.4% rate of infection in 1998, increased to 2.3% in 2004, and 2.9% in 2007, respectively [44].

The infection rate among IBD patients varies in different geographic regions: 3.4% in UC patients and 5.9% in CD patients were reported in the Netherlands [46]; 8.5% in CD and 24.7% in UC were reported in Chinese patients [47]. The rate of CDI among IBD patients with an ileal pouch anal anastomosis (IPAA) varies between 10.7 and 18.3% [48,49].

In a national survey, Nguyen et al. showed that the incidence of CDI in UC almost doubled in 7 years (1998–2004). This was associated with increased hospital stays, healthcare costs, and mortality rates [50]. The increasing presence of CDI in the setting of IBD could be explained by increasing clinical awareness, improvements in diagnostic protocols, and possibly by the occurrence of more virulent strains.

## 3. Pathogenic Mechanisms

*C. difficile* is an anaerobic, spore-forming, Gram-positive bacterium, identified as a causative agent of antibiotic-associated pseudomembranous colitis [51]. Infection comprises a spectrum of clinical manifestations that range from asymptomatic carriage or mild symptoms to life-threatening conditions (toxic megacolon, colonic perforation) and death [52].

Spores are transmitted by the fecal–oral route. Acquisition of C difficile in hospitalized patients occurs by ingestion of spores from an infected patient, transmission of pathogens via contaminated hands of healthcare personnel, or from contaminated surfaces. Antibiotic use affects gut microbiota composition and creates a favorable environment for C difficile colonization.

Virulence factors responsible for damage of intestinal mucosa are two monoglycosyltransferases, enterotoxin A (TcdA) and cytotoxin B (TcdB), released by CD toxigenic strains [53]. These two toxins are encoded by tcdA and tcdB genes, located on the 19.6 kb pathogenicity locus (PaLoc). The PaLoc contains three other genes involved in regulation of the toxin genes (tcdR, tcdC) and toxin secretion (tcdE) [54,55].

Toxin A and toxin B bind to a specific receptor on colonic epithelial cells and affect intestinal barrier integrity by disruption of epithelial tight junctions and promotion of inflammatory host responses by neutrophil recruitment and activation of pro-inflammatory cytokines [56]. Additionally, vascular changes caused by these toxins, with increased production of vascular endothelial growth factor A (VEGF-A) and increased colonic vascular permeability, are involved in disease pathogenesis [57]. Toxigenic strains may produce both toxin A and toxin B (causing the majority of CDIs), TcdB alone (A−B+strains), or TcdA alone (A+B−strains) [58,59,60,61]. Recent research proves the positive correlation between high serum levels of toxin A and the severity of CDI [62]. Toxemia could explain systemic clinical manifestations in severe cases of CDI [63,64]. However, since circulating anti-toxin antibodies prevent the detection of *C. difficile* toxins in blood, current methods used to detect serum toxins levels have a low sensitivity [65,66].

Cytokine profile could also serve as a predictive marker for CDI severity. Yu H et al. demonstrated that high serum levels of IL-2 and IL-15 are associated with severe disease and poor prognosis, while high serum levels of IL-5 and gamma interferon are encountered in less severe disease. Therefore, anti-inflammatory agents could play an important therapeutic role by controlling host response and consequently reducing intestinal injury [67].

A third toxin, named C difficile transferase or binary toxin CDT, also involved in disease severity, increases bacterial adherence to intestinal epithelial cells and uptake of TcdA and TcdB [68]. Of note, 23% of the toxigenic strains, including the hypervirulent ribotype 027 and 078, produce binary toxin [69].

Asymptomatic carriers of toxigenic strains represent a population reservoir that favors the spread of infection [70,71]. *C. difficile* carriage was reported in 8.2% of IBD outpatients in clinical remission [72]. It is possible that the asymptomatic carriage of non-toxigenic strains confers protection against the toxigenic ones [73].

## 4. Risk of CDI in IBD Patients

Risk factors for *Clostridioides difficile* acquisition in IBD patients were analyzed in various and heterogenous studies and some contradictory data were obtained. IBD-related risk factors for CDI (such as disease location, severity and disease extent, and immunosuppressive therapy) were assessed, together with traditional risk factors (age, antibiotic exposure, hospitalization, and comorbidities). IBD patients with superimposed CDI showed distinct features, compared to the general population: younger age, with no recent exposure to antibiotics, presenting more frequently with a community-acquired infection [74].

### 4.1. The Role of Traditional Risk Factors

Increasing age and comorbidities were identified as risk factors for CDI in IBD patients, similarly to the general population [45]. The predisposition of patients with IBD colitis towards *C. difficile* acquisition could be explained by gut microbiota disturbance [75], leading to impaired resistance to *C. difficile*, even in the absence of recent antimicrobial exposure [74]. The reported rate of prior antibiotic exposure as a risk factor for CDI in IBD varies: 42% of IBD patients (compared to 69% of non-IBD patients) were exposed to antibiotics within three months before CDI in one report [42], whereas no prior antibiotic therapy was identified in 39% of CDI-IBD cases in another report [76]. Other authors identified the recent use of antibiotics in IBD as a risk factor for both CDI and for recurrent CDI as well [77]. Exposure to nonsteroidal anti-inflammatory drugs within two months prior to hospital admission for IBD flare was a risk factor for CDI in a study performed by Regnault et al. [78]. Although CDI is mostly a healthcare-associated infection, some authors demonstrated that CDI in IBD is more frequently community-acquired [45]. Maharshak et al. identified recent hospitalization (within the prior two months) and the use of antacids as significant risk factors for CDI in IBD patients [79].

### 4.2. The Role of IBD-Related Risk Factors

The rate of CDI is higher in UC compared to CD [45], and colonic involvement represents a significant risk factor for CD acquisition (OR 3.12; 95% CI: 1.28–5.12) [76]. Patients with pancolitis or left-sided colitis, more than patients with limited colonic involvement, together with patients with severe forms of IBD, are at greater risk for CDI [80,81].

Safety concerns have been raised regarding IBD-specific therapy. In a retrospective study of serious bacterial infections among 10,662 IBD patients undergoing immunomodulating therapy, corticosteroid initiation, with or without concomitant immunomodulators, was associated with a threefold increased risk for CDI compared with other immunosuppressants (RR = 3.4; 1.9–6.1). No significant association was found with infliximab use. Antibiotic therapy in the previous six months was found in 57.2% of patients with CDI [82].

Corticosteroids and biologics (infliximab and adalimumab) were identified as risk factors for CDI in IBD patients, as well as metronidazole, hospitalizations, higher ambulatory care visits, shorter duration of IBD, and a higher number of comorbidities [83]. In a systematic review and meta-analysis, Balram et al. showed that biologic therapy, antibiotic use, and colonic involvement are all risk factors for CDI in IBD [84]. Maintenance therapy with immunomodulators (azathioprine, 6-mercaptopurine, methotrexate) was an independent risk factor for CDI in one report [76], whereas presence of a fistula, antibiotics (metronidazole and cephalosporins), and infliximab (particularly in combination with antibiotics) were identified as risk factors for CDI in another report [85].

Razik et al. showed in a retrospective analysis that IBD patients are 33% more likely to have recurrent CDI compared with the general population. Recent antibiotics, 5-ASA, steroids, and biologic therapy (infliximab), as well as non-ileal CD, were predictors of recurrence [86].

### 4.3. Genetic and Immunologic Susceptibility

Additional risk factors for CDI in IBD were studied for identification of susceptible patients. Connelly et al. showed that interleukin-4 gene-associated single-nucleotide polymorphism rs2243250 was significantly associated with CDI in IBD patients [87].

The susceptibility for recurrent CDI can be explained by altered host humoral immunity, while high-serum anti-toxin A or anti-toxin B antibodies are protective against recurrent infections [88,89]. In older IBD patients, an impaired capacity to produce toxin-specific antibodies and memory B-cell responses could explain the increased risk for CDI [90].

## 5. Diagnosis of CDI in IBD

### 5.1. Clinical Assessment

Common symptoms of CDI are: new onset diarrhea, ranging from 3 loose stools to more than 10 watery stools within a 24 h period, abdominal pain, fever, dehydration, and blood in the stool. Severe complications occur in fulminant disease and can evolve to adynamic ileus, hypotension, shock, toxic megacolon, colonic perforation and peritonitis. In IBD patients who underwent total colectomy, an increased ileostomy output with associated dehydration occurs in acute enteritis caused by *C. difficile*, together with fever and leukocytosis [91]. An increasing number of stools in patients with IPAA should raise the suspicion for *C. difficile* pouchitis [48,49].

### 5.2. Laboratory Tests

Laboratory diagnosis is recommended in symptomatic patients, and consists in the detection of toxins in a liquid stool sample. The culture for toxin-producing *Clostridioides difficile* (toxigenic culture TC) and cell culture cytotoxicity neutralization assay (CCNA) are methods with the highest sensitivity, but these require time and are expensive [92,93]. Compared to these methods, enzyme immunoassay (EIA) for toxins has the advantage of a rapid and easy-to-perform test, but with poor sensitivity [94]. Nucleic acid amplification tests (NAATs) represent sensitive methods to detect the presence of CD toxins genes. However, NAAT alone cannot make the distinction between colonization and infection and may lead to overdiagnosis and unnecessary treatment of asymptomatic carriers [95,96]. The glutamate dehydrogenase (GDH) test is a useful screening test that detects the enzyme GDH, expressed by toxigenic and non-toxigenic CD strains [97]. Additional laboratory tests are white blood cell count, serum albumin, creatinine, and electrolytes. Leukocytosis (>15,000 cells/mL) and increased serum creatinine level (>1.5 mg/dL) are markers of severe CDI. Patients with low albumin levels are at a higher risk of developing severe infection and recurrent CDI [98,99].

A new serological assay has been proposed to predict the risk for recurrent infection: an ELISA test measuring serum levels of toxin-A- and -B-specific antibodies, potentially identifying those patients with low levels of antibodies who are more susceptible to recurrence. However, extended research is needed to evaluate the applicability of this assay in current practice [100].

### 5.3. Diagnostic Algorithm

The diagnostic algorithm for CDI consists in EIA for GDH detection or NAAT, followed by EIA testing for toxins of the stool sample. Another approach is to test the stool sample simultaneously for GDH and free toxins (EIA), followed by NAAT or TC. According to the European Society of Clinical Microbiology and Infectious Diseases (ESCMID), the Infectious Diseases Society of America (IDSA), and the Society for Healthcare Epidemiology of America (SHEA) guidelines, a combination of diagnostic tests is recommended [24,101]. Clinical judgement is advocated in making the distinction between colonization versus infection in PCR positives/toxins negative cases [101]. This approach prevents unnecessary antibiotic use that may cause disruption of normal gut microbiota and selection of vancomycin-resistant enterococcus species [102,103]. A positive GDH test with a negative EIA test for toxins demands immediate treatment in selected cases with severe symptoms and a high index of suspicion for CDI in IBD patients [104].

### 5.4. Endoscopic and Histopathological Evaluation

The classic morphologic features of CDI, consisting in the detection of inflammatory pseudomembranes on endoscopic and histological evaluation, are generally not found in IBD patients. Pseudomembranes represent a layer of fibrin, necrotic epithelial cells, mucus, leukocytes, that create the typical endoscopic appearance of elevated whitish or yellowish plaques on the colonic mucosal surface [76,105]. It is speculated that an altered mucosal inflammatory response in active IBD, or as a result of immunomodulator use, may result in the inability to form pseudomembranes [106].

A lower gastrointestinal endoscopy with biopsies is useful to rule out other potential causes of diarrhea, such as cytomegalovirus colitis, ischemic colitis or amoebiasis [107], and to evaluate the extension and the severity of IBD.

### 5.5. Imagistic Evaluation

Radiologic evaluation (abdominal X rays and computed tomography CT) represents an effective diagnostic method in severe, complicated cases, by detecting loss of haustration, dilated transverse or right colon in toxic megacolon, air-fluid levels, or free intraperitoneal gas in perforation. Imaging provides clues in severe colonic inflammation: bowel wall thickening, thumbprinting, accordion sign, pericolonic stranding and ascites [108,109,110]. Similar findings, consisting of small bowel dilatation, wall thickening, and ascites, have been described in CD enteritis [111,112]. Abdominal ultrasound allows the assessment of disease extent and severity, and specific features for both CDI and IBD can be detected: thickening of the bowel wall, colonic dehaustration, free fluid between bowel loops, colonic dilatation, or free air in complicated cases.

## 6. Impact of CDI on Clinical Outcome

CDI has a negative influence on IBD course, resulting in increased morbidity and mortality [84]. In a national study performed in England over a 6-year period, Jen MH et al. detected an increased morbidity and mortality rate in hospitalized CDI-IBD patients compared to IBD alone patients. Emergency surgery and colectomy rates were increased in the first group, and a longer length of hospital stay was found [43]. Similarly, a three-day increase in hospital stay in IBD-CDI patients was reported by Ananthakrishnan et al., as well as a four times greater mortality rate compared to IBD-alone patients, or CDI-alone patients [3]. Zhang et al. showed an increased length and frequency in hospitalisation. In the 2 years of follow-up after CDI, the rate of bowel surgery increased in IBD patients, and the rate of remission decreased. Patients with CD were particularly affected by CDI in this research, and developed active moderate or severe disease during the follow-up period [85].

The negative impact of CDI on UC course has been demonstrated in many studies: an increased rate of hospitalization, more emergency room visits in the first year after infection [113], requirement for medical therapy escalation, a higher colectomy rate [114], and an increased risk for postoperative complications and death [115] have all been reported.

In a systematic review and meta-analysis, a significant higher risk of surgery was found in UC-CDI patients, compared with UC-alone patients (OR = 1.76, 95% CI = 1.36–2.28), although the severity of underlying IBD and associated comorbidities, that could influence the poor outcomes, had not been assessed [116]. Preoperative *C. difficile* colitis was associated with pouch failure in a study performed by Skowron et al. [117]. Murthy et al. showed the negative long-term impact of CDI in hospitalized UC patients, who presented a higher adjusted 5-year risk of mortality (adjusted hazard ratio 2.40, 95% CI 1.37–4.20) than patients without *C. difficile* colitis [118].

## 7. Treatment Strategies

### 7.1. Prevention

Preventative strategies remain an important step in disease control. Rigorous in-hospital measures consisting of protective equipment use (gloves and gowns), cleaning and disinfection protocols (handwashing, disinfection of surfaces), isolation measures, and antibiotic stewardship, are all critical. Chlorine-based disinfectants are effective to eradicate spores, whilst alcohol and ammonium-based detergents have no sporicidal action [119].

### 7.2. Treatment of CDI in the Setting of IBD

The first line of treatment in CDI consists of oral vancomycin 125 mg 4 times a day for 10 days [24]. A reduced necessity for colectomy was reported after vancomycin use in IBD patients with superimposed CDI. However, the rate of CDI recurrence after vancomycin requires additional therapy [76]. Fidaxomicin 200 mg twice daily for 10 days is a useful alternative for the treatment of an initial CDI episode [24]. Oral metronidazole 500 mg 3 times a day for 10 days could also be administrated, but therapeutic response is poor [120].

In severe, complicated and fulminant disease, oral vancomycin 500 mg 4 times a day and enemas containing 500 mg vancomycin every 6 h are recommended, together with intravenously metronidazole 500 mg every 8 h [24]. Aggressive resuscitation, consisting of intravenous fluid replacement, albumin and electrolyte supplementation, and early surgical consultation, are mandatory in patients with fulminant colitis. A subtotal colectomy with end ileostomy is indicated in severe-complicated or fulminant colitis, or alternatively, a diverting loop ileostomy with colonic lavage [121]. A brief summary of the therapeutic approach in patients with IBD flare-up is presented in Figure 1.

### 7.3. Treatment of Recurrent Infection

Recurrent CDI or CDI refractory to the standard oral vancomycin regimen are encountered in patients with UC and CD [76], including patients with IPAA [48,122]. For the first recurrence of CDI, vancomycin in standard doses is recommended if the first episode was previously treated with metronidazole. After a first episode treated with vancomycin, a tapered and pulsed vancomycin regimen (125 mg 4 times per day for 10–14 days, then 125 mg 2 times per day for a week, once per day for a week, 125 mg every 2 or 3 days for 2–8 weeks) or fidaxomicin is further recommended.

A tapered and pulsed vancomycin regimen, vancomycin in standard dosage 10 days followed by rifaximin 400 mg TID for 20 days, fidaxomicin 200 mg BID for 10 days, and vancomicin or fidaxomicin followed by fecal microbiota transplantation (FMT), represent therapeutic choices in recurrent infection [24,123]. Data supporting the efficacy of prolonged oral vancomycin therapy in IBD result from a study performed by Lei et al., who found lower rates of CDI recurrence as a result of long-duration therapy (21–42 days), comparing with short-duration vancomycin therapy (10–14 days) [124].

New therapies for restoration of the gut microbiota are proposed to prevent CDI recurrences and to suppress antibiotic-resistant organisms (such as vancomycin-resistant enterococci) [125]. FMT proved efficient in refractory and recurrent CDI [126,127], including in the setting of IBD, although concerns regarding safety and efficacy have been raised in IBD patients [128]. According to some authors, the results of FMT for CDI seem to be less effective in IBD patients, with a lower rate of infection clearing, when compared with non-IBD patients (74.4% in IBD versus 92.1% in non-IBD patients). Furthermore, an IBD exacerbation occurred in greater than 25% of patients after a single colonoscopic FMT [129].

However, other reports showed promising results for FMT in IBD patients with concurrent CDI. Tariq et al. reported that FMT was efficient in treating recurrent CDI in IBD, with an overall cure rate of 80% [130]. In another cohort study comprising 67 IBD patients with recurrent CDI, a cure was obtained in 79% of patients, with an improvement in IBD activity in 37% of patients. IBD flares requiring hospitalization were reported in a minority of patients (2.9%) [131]. In a prospective trial, FMT was performed in 49 IBD patients with two or more CDI episodes, and IBD improvement was obtained in 73.3% of CD patients and in 62% of UC patients [132]. In a systematic review and meta-analysis regarding the risk of IBD flares after FMT, the pooled rate of IBD worsening was 22.7% (95% CI: 13–36%) after FMT for CDI treatment, compared to 11.1% (95% CI: 7–17%) after FMT for IBD treatment. When high-quality studies and randomized controls trials (RCTS) were analyzed, a marginal risk for IBD worsening following FMT (4.6% (95% CI: 1.8–11%)) was detected. This highlights the need for standardized research protocols in this field [133]. Concerning the long-term benefit of FMT, no differences were founded regarding the risk for CDI recurrence at six months after FMT between IBD and non-IBD patients [134].

### 7.4. Therapeutic Alternatives

Facing the emergence of hypervirulent strains with higher resistance to traditional treatments, as well as higher morbidity and mortality rates, novel therapies that target virulence factors have been developed [135]. Actoxumab and Bezlotoxumab are monoclonal antibodies directed against toxins A and B, respectively. Bezlotoxumab is a humanized monoclonal antibody, approved by the FDA for adult patients with a high risk of recurrent infection, that binds to toxin B and prevents CDI recurrence [6,136]. A total number of 44 patients with IBD were included in MODIFY I/II Phase 3 trials. When compared with non-IBD patients, IBD patients included in the study were younger, more frequently outpatients, with immunosuppression, and with less antibiotic exposure. Bezlotoxumab proved to be efficient in reducing the incidence of recurrent CDI in IBD patients [137]. Recent guidelines included Bezlotoxumab 10 mg/kg infusion as an adjunct treatment in high-risk patients, for initial or recurrent severe CDI [138].

A tetravaccine composed of attenuated forms of TcdA, TcdB and binary toxin (CDTa and CDTb) has been developed by Secore et al., with increased efficacy against hypervirulent NAP1 strains [139].

Probiotics proved the benefit in reducing the risk of CDI in patients who receive antibiotics, although the optimal dose and strain should be determined. [140,141]. According to international guidelines, there are limited data to support the use of probiotics as an adjunctive therapy to standard antibiotics (vancomycin) for the first episode of CDI [121]. The benefit of probiotics in combination with vancomycin or metronidazole for prevention of CDI recurrence has been evaluated in various research studies [142,143].

According to some authors, probiotics may cause a delayed microbiome reconstitution after antibiotic therapy [144]. Allegretti et al. showed a greater risk of CDI recurrence in patients who received probiotics prophylaxis after FMT for recurrent CDI [145]. Additional research is required to evaluate the efficacy of probiotics in IBD patients with superimposed CDI.

### 7.5. Treatment of Underlying IBD in CDI

Concerning therapies prescribed to control IBD flares, published data show the negative impact of immunosuppressive therapy on clinical outcomes, although there are contradictory results from other studies. Ben-Horin et al. reported worse outcomes (death or colectomy within 3 months of admission, in-hospital megacolon, bowel perforation, hemodynamic shock, or respiratory failure) in IBD-CDI patients treated with antibiotics and immunomodulators (20 mg of prednisone or greater, thiopurines, methotrexate, cyclosporine, tacrolimus, or biologics), when compared with patients treated with antibiotics alone [146]. A combination of two or three immunomodulators was associated with negative outcomes on multivariate analysis [146]. In a study performed by Lim et al., corticosteroids use following CDI diagnosis by PCR in UC-hospitalized patients was associated with an increased length of hospitalization, higher ICU admission and colectomy rate [147]. In another retrospective analysis, corticosteroid escalation in CDI-IBD was associated with a higher risk of colon surgery within one year of CDI, while immunomodulator and biologic therapy changes did not influence the outcomes [148]. According to other researchers, older age and altered laboratory parameters (serum albumin level < 3 g/dL, hemoglobin < 9 g/dL, and serum creatinine > 1.5 g/dL) were identified as predictors of severe outcomes, and no association between IBD-specific therapy (immunosuppressive therapy or steroids) and the risk of colectomy or death within 180 days of first diagnosis of CDI was found [149].

The best therapeutic approach remains under debate, and must be guided by the evolution and characteristics of the individual case. In the current practice, strategies vary among gastroenterologists, as was seen in the results from a questionnaire regarding therapeutic choices in UC with concurrent CDI: 46% of participants recommended immunosuppressive drugs in combination with antibiotics, whilst 54% recommended antibiotic alone. Only 11% of participants, most of them IBD experts, agreed to stop therapy with azathioprine [150].

According to AGA recommendations, early specific antibiotic therapy is warranted, while corticosteroid or immunosuppressive escalation should be postponed, although no definitive indications to withhold, maintain, or escalate immunosuppressive therapy can be made in the absence of prospective data [74]. In cases with persistent severe symptoms suggesting an exacerbation of underlying IBD, with no clinical improvement within two to three days of vancomycin therapy, corticosteroid initiation or even immunosuppressive therapy escalation should be considered, in concert with rigorous monitoring for possible complications [74,151].

In a multicenter retrospective cohort study including 207 CDI-IBD patients, therapy escalation with corticosteroids or biologics after the initiation of specific antibiotics for CD was not associated with severe outcomes (including death, sepsis, or colectomy at 90 days), which supports the current therapeutic recommendations. Hypoalbuminemia, advanced age and comorbidities (cardiovascular, renal, and rheumatologic disease) were correlated with severe outcomes [152]. Similarly, no association between early corticosteroids exposure (within 48 h from admission) and negative outcomes (colectomy and mortality rate at 1 year) has been detected in a recent retrospective study of IBD patients with CDI [153].

## 8. Conclusions

Clostridoides difficile infection in the setting of IBD represents a genuine challenge for the practician. Symptoms mimicking an IBD exacerbation should raise the suspicion of a concurrent infection, and initiate the pursuit of an early diagnosis and appropriate therapy. There are ongoing debates concerning the risk factors for CDI and the predictors for negative outcomes in IBD, with heterogenous data from different analyses. CDI has a proven negative impact on the course of IBD, leading to significant morbidity and mortality. Progress has been made in the fields of diagnostic techniques and therapies, but larger prospective studies are required to determine best management strategies and standardized protocols in complicated, refractory or recurrent CDI in IBD patients.

## Figures and Tables

**Figure 1 pathogens-11-00819-f001:**
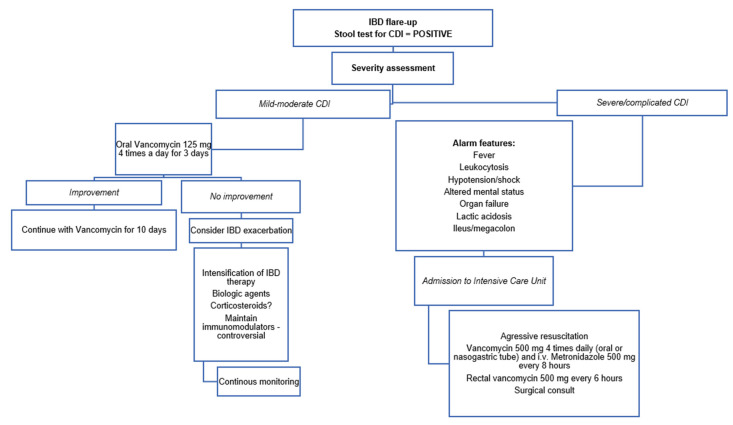
Therapeutic approach to the patient with inflammatory bowel diseases (IBD) and *Clostridioides difficile* infection (CDI).

## Data Availability

Not applicable.

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
