# Peer review of "The Current Knowledge on *Clostridioides"

_pathogens, 2022, doi:10.3390/pathogens11070819_

Round 1

Reviewer 1 Report

In manuscript " The current knowledge on Clostridium Difficile Infection in 2 patients with Inflammatory Bowel Diseases" authors described the current evidences in the field of the CDI in patients with inflammatory bowel disease, pointing to pathogenic mechanisms, risk factors for infection, diagnostic steps, clinical impact and outcomes, and specific management.

The content of the review paper is interesting and important due to the increasing incidence of C. difficile infections in hospitalized population and it is the patients with IBD who are the vulnerable population. The reason for this is the coincidence of symptoms, which leads to the later detection of CDI and consequent complications.

Topic is up to date and original.

The diagnostics that are a great challenge in patients with IBD and CDI, as well as the therapy with recommendations, are described in detail.

As a microbiologist, I am bothered by incorrectly written names of bacteria and that part needs to be corrected from the title throughout the text. The part related to therapy and the clinical picture is understandably written.

The article is easy to read only that you need to write the names of the bacteria and the abbreviation of the infection correctly.

I think the conclusion is in line with the issues described in the manuscript.

My main issue is the misspelled names of bacteria and the abbreviation infection. The incorrect spelling of the name of the bacterium Clostridium difficile because its correct name is Clostridioides difficile and so it should be stated. Not the other way around. Also through the text it is written C difficile or C. difficile in italic and without it and it all needs to be balanced and corrected. A scheme or drawing would be welcome when describing the pathogenesis.

Reviewer 2 Report

The work is a good overview, well written and comprehensively described

Author Response

Dear Editor,

Thank you for giving us the opportunity to submit a revised draft of the manuscript “The current knowledge on Clostridoides Difficile Infection in patients with Inflammatory Bowel Diseases” for publication in the Pathogens – special Issue number. We appreciate the time and effort that you and the reviewers dedicated to providing feedback on our manuscript and are grateful for the insightful comments on and valuable improvements to our paper

Reviewer 3 Report

The main concern on this review is limited contribution to the clinical practice and science. Recently there has been published a similar review paper Dalal RS, Allegretti JR. Diagnosis and management of Clostridioides difficile infection in patients with inflammatory bowel disease. Curr Opin Gastroenterol. 2021 Jul 1;37(4):336-343. doi: 10.1097/MOG.0000000000000739.  

The authors should highlight and rationale the need of another review about C. difficile infection in IBD. What has changed since the paper by Dalal et al.?

The summary of scientific evidence (i.e. summary of available meta analyses) for treatment optoins should be added. What is the role of probiotics in prevention of CDI in aptients with IBD?

 The diagnostic algorithm for CDI as a figure would be valuable.

"Clostridioides" instead of "Clostridium" should be in the title and main text.
